# Experimental Investigation of the Wear Behaviour of Coated Polymer Gears

**DOI:** 10.3390/polym13203588

**Published:** 2021-10-18

**Authors:** Brigita Polanec, Franc Zupanič, Tonica Bončina, Frančišek Tašner, Srečko Glodež

**Affiliations:** Faculty of Mechanical Engineering, University of Maribor, Smetanova 17, 2000 Maribor, Slovenia; brigita.polanec@um.si (B.P.); franc.zupanic@um.si (F.Z.); tonica.boncina@um.si (T.B.); francisek.tasner@um.si (F.T.)

**Keywords:** polymer gears, coatings, experimental testing, wear

## Abstract

A comprehensive experimental investigation of the wear behaviour of coated spur polymer gears made of POM is performed in this study. The three physical vapour deposition (PVD) coatings investigated were aluminium (Al), chromium (Cr), and chromium nitrite (CrN). Al was deposited in three process steps: By plasma activation, metallisation of Al by the magnetron sputtering process, and by plasma polymerisation. Cr deposition was performed in only one step, namely, the metallization of Cr by the magnetron sputtering process. The deposition of CrN was carried out in two steps: the first involved the metallization of Cr by the magnetron sputtering process while the second step, vapour deposition, involved the reactive metallisation of Cr with nitrogen, also by the magnetron sputtering process. The gears were tested on an in-house developed testing rig for different torques (16, 20, 24 and 30 Nm) and rotational speed of 1000 rpm. The duration of the experiments was set to 13 h, when the tooth thickness, and, consequently, the wear of the tooth flank was recorded. The experimental results showed that the influence of metallisation with aluminium, chromium, and chromium nitrite surface coatings on the wear behaviour of the analysed polymer gear is not significant. This is probably due to the fact that the analysed coatings were, in all cases, very thin (less than 500 nm), and therefore did not influence the wear resistance significantly. In that respect, an additional testing using thicker coatings should be applied in the further research work.

## 1. Introduction

Polymer gears are used widely in many engineering applications, such as office appliances, mechatronic devices, household facilities, computer and laboratory equipment, medical instruments, etc. [1,2,3,4]. These polymer gears can be produced by classical cutting processes or, for large series production, by injection moulding [5,6]. Some of the main benefits of polymer gears are high size-weight ratio, low coefficient of friction, self-lubrication, high resistance against impact loading, ability to absorb and damp vibration, reduced noise, ability to be used in food preparation areas, etc. [7,8,9,10]. However, polymer gears also have some disadvantages, such as less load carrying capacity and lower operating temperatures if compared to the metal gears, difficulties in achieving high tolerances (especially in the case of moulded gears), relatively high dimensional variations due to temperature and humidity conditions, etc. [11,12,13]. In order to improve the polymer gear characteristics regarding heat resistance and higher strength, glass fibre-reinforced polymer gears have been used increasingly in recent years [14,15,16]. 

Because gears are key machine elements in many engineering applications, the proper estimation of load capacity against failures under given loading conditions is crucial when dimensioning the gear drives. In the case of polymer gears, the standardised procedure according to the VDI 2736 [17] is usually used for that purpose. Furthermore, the following failure types of polymer gears are addressed in [17] and explained additionally in [18,19]: melting, tooth root fracture, tooth flank fracture, pitting, tooth wear and tooth deformation. 

Recently, many researchers have been investigating different types of polymer gears in respect to the failure modes as described above. Because polymer gears usually run in dry operating conditions (without lubrication), the high contact friction and, consequently, high degree of wear, may be the main reason for the short service life of a gear drive, especially in medium to high power transmission applications. Singh et al. [20] investigated the wear behaviour of polymer gears made of ABS, HDPE and POM at torque levels 0.8 to 2.0 Nm and rotational speeds 600 to 1200 rpm. Their results showed that the specific wear rate is maximum for ABS and minimum for POM. Li et al. [21] studied the wear behaviour of polymer gears with consideration of the engagement ratio of gear flanks. The experimental results have shown that significant reduction of wear can be achieved with the micro geometry modification (tip relief) of gear flanks. Evans et al. [22] proposed a novel wear mechanism of POM-gears which enables prediction of the volume of removed material for a given load and speed of the analysed gear pair. The dynamic interaction between contact loads and tooth wear of meshing polymer gears was also studied by Lin et al. [23]. Their numerical analyses have shown that the dynamic load histogram of an engaged polymer gear pair can influence the tooth wear significantly. Mao et al. [24] have been investigating the influence of the manufacturing process on the wear behaviour of polymer gears. Their results have shown that the wear rate is independent of the manufacturing process (machine cutting, injection moulding), which leads to the conclusion that the machine cut polymer gears can be designed using the existing methods for injection moulded polymer gears.

Polymer gears are often used in applications in which lubricants cannot be used, such as food processing machines and office equipment [25]. However, high running temperatures may lead to the short operating life of polymer gears, especially in medium to high power transmission applications [26]. An often-used method to reduce friction of meshing gear pairs made of polymers is the applying of low-frictional coatings [27], which may also improve the surface properties of teeth flanks [28], and, consequently, lead to the higher efficiency of the meshing gear pair [29]. The experimental work by Dearn et al. [30] presented an attempt to control friction and wear by reducing the running temperatures by using a series of solid lubricant coatings (molybdenum disulphide-MoS2, graphite flake, boron nitride and poly-tetra-fluoro-ethylene (PTFE)) deposited on the polymer gear teeth flanks. The experimental results indicated that the PTFE-coating provided the greatest reduction of friction and wear for the analysed polymer gears. Furthermore, Bae et al. [31] proposed an FEM-model to investigate the contact stress response of a coated polymer gear with the frictional effect during gear operation. Their numerical results have shown that a coating of 2 μm thickness had a negligible effect on the contact stress during the mesh cycle, because the thickness was insufficient to affect the bulk deformation behaviour of the polymer gears. Similar conclusions may also be found in the experimental study proposed by Petrov et al. [32].

Physical vapour deposition (PVD) is another technique which may be used to improve the wear resistance of contacting mechanical elements. As presented by Baptista et al. [33], the PVD-technology is used widely for the deposition of thin films to improve tribological behaviour, optical enhancement [34,35], visual/aesthetical upgrading, etc. PVD processes allow the deposition of mono-layered or multi-layered coatings, as well as different alloy compositions [36,37]. The authors in [38] also investigated the influence of different metal coatings on the mechanical and physical properties of carbon fibre reinforced polymers (CFRPs) which have been used increasingly in the aerospace and automotive industries. Maurer et al. [39] showed that multilayer thin films of pure Ti or Ti/TiN deposited on epoxy and PEEK based CFRP increased their erosion resistant significantly. Similar conclusions have also been made by Coto et al. [40], who investigated the role of surface finishing and an interfacial lacquer layer on the particle erosion mechanisms of Ti/TiN multilayer PVD coatings on epoxy-based CFRP.

The most commonly used PVD-method is magnetron sputtering. This is a process of material vaporisation by bombarding the target material with high-energy ions. The process takes place in a vacuum chamber containing an inert gas, the substrate, and the coating material [33,41]. Magnetron sputtering uses a static magnetic field placed on the cathode and parallel to the cathode surface. The magnetic field allows condensation of the plasma in front of the target, lower electric current to the substrate, and, thus, lower heating. Compared to a conventional discharge, the magnetic field keeps the electrons near the cathode surface as long as possible, where they enhance ionisation. This leads to the formation of a dense plasma. The plasma is the source of the ions with which we sprinkle the target. The rate of sputtering depends on three factors: The atomic mass of the ions, the flux density of the ions and the energy of the ions (Figure 1). With this method, it is possible to prepare hard nanocomposite and multicomponent coatings. Most of the magnetic forces are confined in the space in front of the target, while the rest of the forces extend into the space against the substrates. Such a magnetic field keeps the electrons in the space in front of the substrate for a longer time, where they ionise. With a negative electric potential the energy of the ions increases. When ions collide with a substrate, their energy is transferred to several coating atoms. However, this energy has a major impact on the physicochemical properties of the coatings thus formed. The more energy the atoms of the coating receive, the better their adhesion to the substrate, the microstructure of the coating is more compact, and the internal stresses are higher [42,43].

To improve the final characteristics of PVD-coatings, some additional technologies are often used with combinations of magnetron sputtering [44]: plasma activation, plasma polymerisation, etc. Namely, polymers have a low surface energy which leads to poorer adhesion of the material. To improve adhesion, the polymer surface must be activated before the coating process. In plasma activation, various interactions occur with the polymer surface: First, the incident ions cause the desorption of impurity molecules and the formation of radicals. Secondly, electrons from the plasma can trigger the disintegration of molecules, or even the entanglement of polymer chains, and thirdly, the radicals of the gas molecules react at the surface of the polymer, increasing its reactivity, and, thus, the surface energy. When using the plasma activation process, a thin layer should be applied immediately. Otherwise, the surfaces would become nonpolar again, meaning that the process would have to be repeated. On the other hand, plasma polymerisation is a process for producing protective coatings and various thin films with engineering applications. In plasma polymerisation, a portion of hydrocarbon, fluorocarbon and organic molecules are deposited on the surface of the substrate in the presence of oxygen, nitrogen, or silicon, to form a polymer layer. The advantages of plasma polymerisation over conventional polymerisation are that the polymer layers can be made from almost any material that can be gasified, that such polymers are highly cross-linked, making them insoluble and impermeable to gases and liquids, that they have high temperature resistance and that they are suitable for making very thin films (nm) that adhere well to most substrates.

An experimental investigation of the wear behaviour of PVD-coated spur polymer gears made of POM is performed in this study. Three PVD-coatings investigated were aluminium (Al), chromium (Cr), and chromium nitrite (CrN). The deposition process of the analysed coatings is described briefly in Section 2.1, while the testing procedure is discussed in Section 2.2.

## 2. Materials and Methods

### 2.1. Deposition Process

In the proposed experimental study, three different coatings were prepared on the polymer gears made of POM. Aluminium (Al) coating was applied through a plasma activation process, followed by metallisation of the aluminium through a magnetron sputtering process, and, finally, a plasma polymerisation step. Chromium metallisation by the magnetron sputtering process was used for the chromium (Cr) coating. The third coating of chromium nitride (CrN) was prepared in two steps, namely, the metallisation of chromium by the magnetron sputtering process, and, finally, the step of reactive metallisation of chromium and nitrogen by the magnetron sputtering process. The process parameters of all three analysed coatings are shown in Table 1.

### 2.2. Experimental Procedure

#### 2.2.1. Sample Preparation

The polymer gear specimens made of POM were machine cut from extruded bars using a hobbing process (the basic parameters of the gears are presented in Table 2). Some of the POM-gears were then coated with Al, Cr or CrN, as shown in Figure 2 and already described in Section 2.1. During the experimental testing (see Section 2.2.3), the tested pinion made of POM was meshed with support gear made of steel.

Due to very thin coatings, indentation tests were used for the determination of hardness and indentation modulus. The equipment used in this study was a Nano Test Vantage (Micro Materials Limited, Wrexham, UK), which was equipped with a Berkovich diamond indenter. The indentation instrument was controlled by an electromagnetic drive loading system with a high-precision coil and a permanent magnet. The tests were carried out with increasing loads. One series of tests was carried from 1 to 10 mN, and the other from 10 to 100 mN. In both cases, the loading time, unloading time and time at the maximum load were 10 s. Table 3 gives the indentation hardness and modulus for uncoated and coated samples. The results show a very small increase of both properties due to the coatings. Figure 3 depicts the variation of hardness at very small loads (each point represents the average value of 5 measurements). The indentation size effect for POM is very small, because hardness does not increase strongly with decreasing the maximum load. The hardness of coated POM was highest at the smallest loads and decreased rapidly with the load. The hardness values are much lower than the known hardness of coated materials, especially Cr and CrN.

#### 2.2.2. Characterisation of the Coatings by Scanning Electron Microscopy

A precision saw, Buehler Isomet 1000 (Lake Bluff, IL, USA), and a diamond blade designed for polymeric materials were used to cut the gear samples. The samples were cut at the edge of the gears in the cross-section and longitudinal directions. For the microscopy of the uncoated polymer substrate, we used the Environmental Scanning Electron Microscope (Quanta 200 3D, FEI, Eindhoven, The Netherlands), that allows analysing the electrically and thermally non-conductive materials. We used two detectors, an LFD (Large Field Detector) for secondary electrons and gaseous BSE for backscattered electrons. The pressure in the chamber was 60 Pa. For the surface and cross-section of the coated samples, we used high resolution scanning electron microscopy (Sirion 400 NC, FEI, Eindhoven, Netherlands), equipped with an energy dispersive spectrometer (INCA x-sight, Oxford Analytical, Bicester, UK).

Figure 4 shows the SEM micrograph of the POM-gear without coating at Site 1 (gear tip) and Site 2 (gear tooth root). The SEM micrograph of the coated surface of the POM-gear is shown in Figure 5, while Figure 6 shows the SEM micrograph of the cross-sections of the POM-gears for all three analysed surface coatings. Here, the presence of thin layers was checked by microchemical EDS-analysis. It is clear that the layers were, in all cases, very thin (less than 50 nm). 

#### 2.2.3. Experimental Testing

The gears were tested on an in-house developed testing rig, as shown in Figure 7. The test rig consists of two rigid steel blocks, which are connected firmly with two connecting bars; together, they form the rigid frame of the whole construction. The closed-loop consists primarily of two operating shafts connected with two gear pairs. One gear pair was used for running the test rig only (both gears were made of steel), the other was a tested one (the tested gear was made of POM and the supported gear was made of steel). The torque was applied with a plain digital torque wrench through the gear with a wrench gap at the accessories for working torque, which consisted of a one-way Clutch Bearing CSK 35 to avoid the back rotation of the shaft. Once the desired torque was applied, the clutch was closed, and the tightening device could be removed.

Before the test began, the tested gears were weighed on a technical balance, Mettler Toledo AX 204 SI 01 05 02, with a weighing accuracy up to 0.1 mg. Furthermore, the tooth thickness was measured along the entire circumference of the gear using a Mitutoyo Absolute dial gage with a roller of diameter 5 mm (see Figure 8). During the test, the temperature was monitored with a thermocouple connected to the Ebro TFI 550 temperature device with the measuring accuracy of 1 °C. The obtained temperature was only used to compare the heating rate during the experimental testing, and did not represented the actual temperature of the gear. The rotational speed was set to 1000 rpm, and was controlled using a Voltcraft DT-10L strobe. The experimental testing was performed for different torques (16, 20, 24 and 30 Nm) and a rotational speed of 1000 rpm. The duration of the experiments was set to 13 h, when the tooth thickness, and, consequently, the wear of the tooth flank was recorded. Up to five tests for each loading configuration were then considered when presenting the experimental results (see Section 3).

## 3. Results and Discussion

### 3.1. Coating Morphology and Thickness

The previous characterisation of the morphology and thickness of the coating was performed by SEM (see Section 2.2.1 and Section 2.2.2). The coatings’ thicknesses were estimated using indentation experiments with a very low indentation rate (10^−4^ mN s^−1^) up to 200 nm of the indentation depth. A sharp change occurred on the indentation curves during loading, when an indenter penetrated through the coating and entered into the substrate. Figure 9 shows the increase of the load during indentations with a low indentation rate. The load increases rapidly when an indenter touches the harder coating surface. As the indenter penetrates through the coating and enters into the POM-substrate, the increase of load with increasing the depth becomes much slower. The depth, where a knee appears on the loading curve, indicates the coating thickness approximately. It was estimated that the thicknesses of the Al, Cr and CrN coatings were 40 ± 5 nm, 24 ± 5 nm and 8 ± 3 nm, respectively.

It is clear that the obtained coating thicknesses were in the range between 8 to 40 nm. This is in good agreement with the experimental studies by Baptista et al. [34] and Ferreira et al. [35], who investigated the multilayer Cr PVD-coating on the polymeric substrate. In their work, the thickness of 25 nm was obtained for each deposited layer. Furthermore, Sing et al. [45] reported the thickness variation of Al-coating on a polycarbonate substrate between 12 and 69 nm, depending on the different parameters of the DC magnetron sputtering process used in their study. Nevertheless, Pedrosa et al. [46] obtained coating thicknesses up to 35 nm for CrN thin films deposited on plasma-activated ABS by reactive magnetron sputtering. Beside the thickness measurements, some authors also evaluated the surface roughness of the coated surface. However, as explained by Baptista et al. [34], the surface roughness decreases according to the increase in the number of layers. This is due to the PVD-process, where the deposition is made preferably in the valleys, attenuating the difference of height between peaks and valleys, thus reducing the roughness. Thus, the surface roughness of coated POM-gears will be analysed in our further investigations considering the multilayer PVD-coatings on POM-gears.

### 3.2. Wear Evaluation

Initially, the POM-polymer gears without coating were tested at torques 16, 20, 24 and 30 Nm. The experimental results are shown in Figure 10. As expected, the wear of the polymer gear increased with increasing of the torque.

Furthermore, all three groups of coated polymer gears (Al, Cr and CrN coatings) were tested at the torque 16 Nm. It is clear from Figure 11 that the influence of the Al-coating on the wear behaviour was very small and can be neglected. A little more beneficial effect can be observed for the Cr- and CrN-coatings.

In the subsequent experimental testing, the Al- and Cr-coated polymer gears were tested at the torque 20 Nm. The experimental results shown in Figure 12 indicate that the influence of the Al- and Cr-coating was quite similar and did not represent a significant improvement of the wear behaviour of the tested gears. Finally, some Al-coated polymer gears were also tested at the torque 24 Nm (see Figure 13) where similar findings were obtained, as already presented above. The experimental results have also shown that, for all three groups of coated POM-gears, the surface coating was removed at a very early stage of the experimental testing. This was probably due to the fact that the analysed coatings (Al, Cr and CrN) were, in all cases, very thin (less than 50 nm), and therefore did not influence the wear resistance significantly. In that respect, additional testing should be performed in the future using multilayer coatings of POM-gears.

Figure 14 shows a photograph of the worn contact surfaces after testing. Because the surface coating was removed in a very early stage of the gear operation, the subsequent wear behaviour may be explained using the known theory valid for uncoated gears. As presented in [17,18,19], the wear is a typical surface failure of non-lubricated (dry) gear pairs, which was already the case in our study. Furthermore, there are both sliding and rolling motions present in the contact of the meshing gear teeth, which, besides the external loading of the gear pair (torque), contributes significantly to the wear of the contacting surfaces. 

It is obvious from Figure 14 that the coatings were removed almost completely due to insufficient adherence of the very thin coatings to the gear surface. As has already been mentioned above the surface coatings were removed in a very short time, which can be attributed to the poor adhesion of coatings to the POM material, or to the inappropriate use of coatings on polymer gears. This is contrary to the experimental study presented in [34,35], where a relatively good adhesion was obtained of a chromium PVD-coating on the polymeric substrate. For that reason, the further investigations will need to analyse the adhesion of coated POM-gears, and also discover the abrasive behaviour of coated surfaces using the available test procedures (i.e., pin-on-disc test, ball-cratering test, etc. [34]).

## 4. Conclusions

An experimental investigation of the wear behaviour of coated spur polymer gears made of POM was performed in the presented study. Three different coatings were prepared and analysed regarding the possible wear reduction of meshing gears: aluminium (Al) coating, chromium (Cr) coating and chromium nitride (CrN) coating. Based on the obtained experimental results, the following conclusions can be made:In general, the influence of the analysed metal coatings on the wear behaviour of POM polymer spur gears is small and does not reduce the wear significantly. Namely, the thickness of the analysed coatings was, in all cases, very thin (less than 500 nm), and did not influence the wear behaviour significantly.If we compare the three analysed coatings, the Cr- and CrN-coatings have a little more beneficial effect compared to the Al-coating.The further study should be focused on the wear evaluation of coated polymer gears, where multilayer (thicker) Al, Cr and CrN coatings will be considered. Furthermore, the additional scratch tests should be performed for the appropriate adhesion evaluation of the analysed coatings. Therefore, more accurate conclusions could be made as to whether metal coatings reduce the wear of POM gears.The further study should also have to consider the measurement of the coefficient of friction for all three analysed surface coatings. Based on such measurements, the tribological behaviour could be analysed and evaluated critically for both dry and lubricated contact of meshing gears.

## Figures and Tables

**Figure 1 polymers-13-03588-f001:**
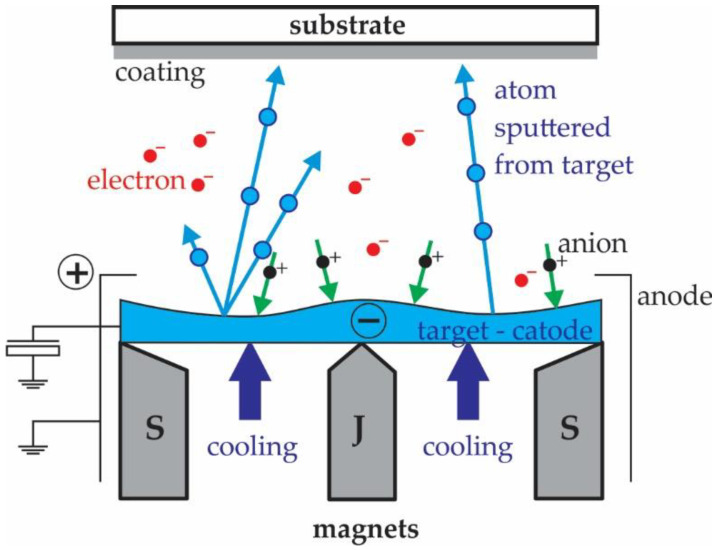
The principle of magnetron sputtering [41].

**Figure 2 polymers-13-03588-f002:**
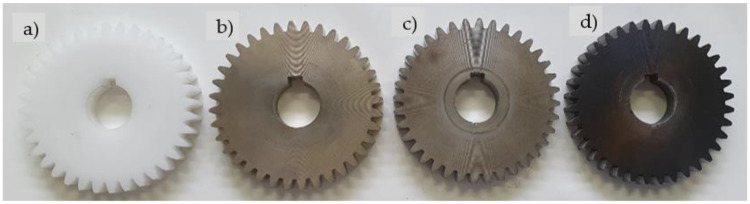
Tested gears made of POM: (**a**) without coating, (**b**) Al-coating, (**c**) Cr-coating, (**d**) CrN- coating.

**Figure 3 polymers-13-03588-f003:**
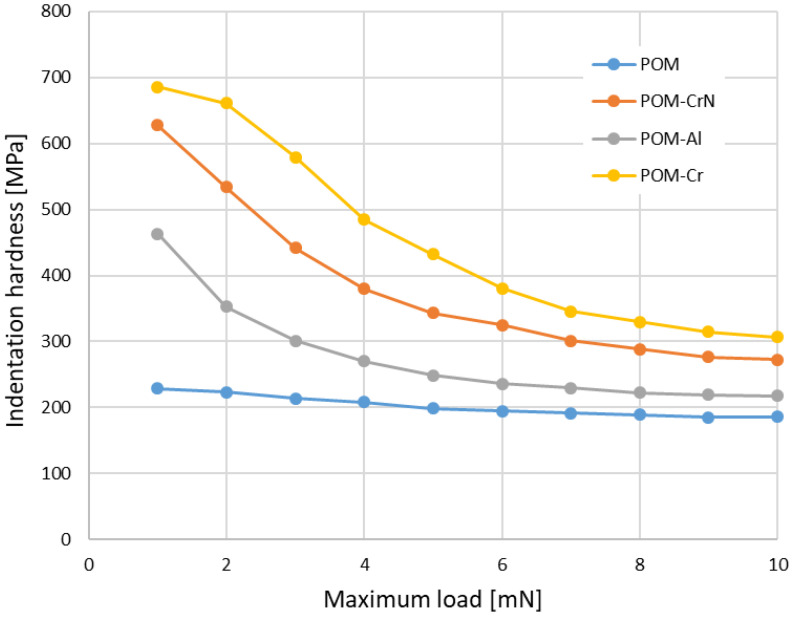
Indentation hardness of POM and coated POM in dependence of the maximum load during indentation experiments.

**Figure 4 polymers-13-03588-f004:**

SEM micrograph of the POM-gear without coating (**a**), SEM micrograph at site 1 (**b**), SEM LFD micrograph at site 2 (**c**).

**Figure 5 polymers-13-03588-f005:**
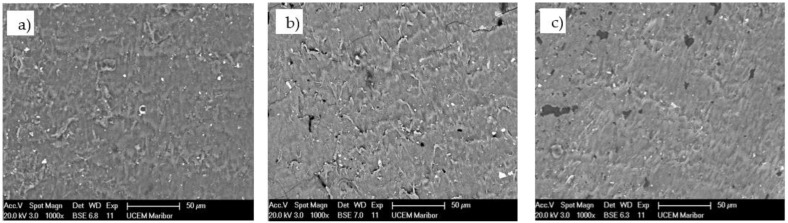
SEM micrograph of the coated surface of POM-gear: (**a**) Al-coating, (**b**) Cr-coating, (**c**) CrN coating.

**Figure 6 polymers-13-03588-f006:**
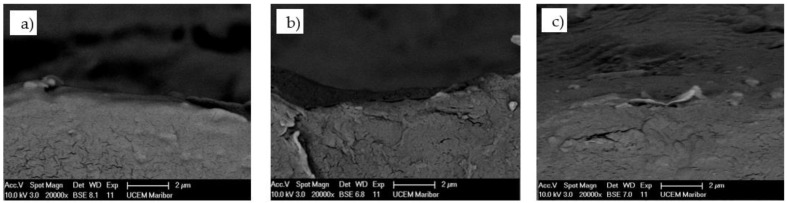
SEM micrograph of the coated cross section of POM-gear: (**a**) Al-coating, (**b**) Cr-coating, (**c**) CrN coating.

**Figure 7 polymers-13-03588-f007:**
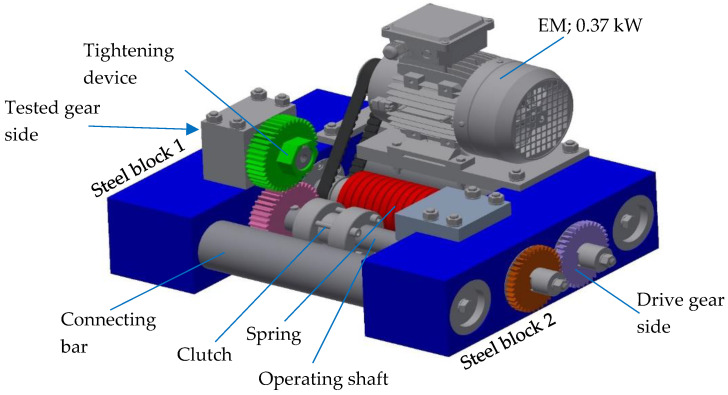
The principle of the testing rig.

**Figure 8 polymers-13-03588-f008:**
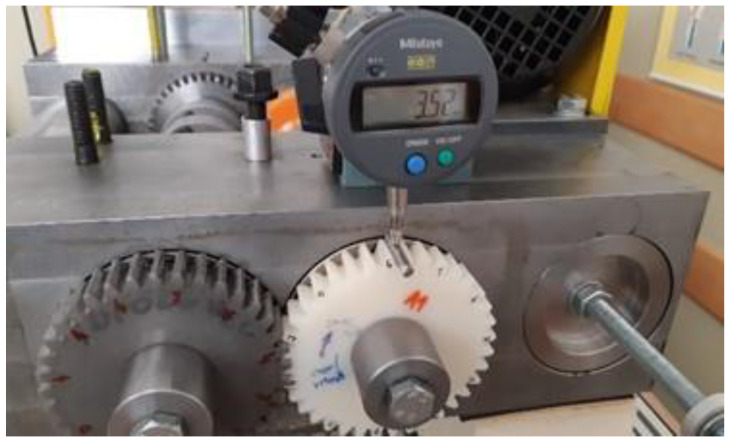
Measurement of the tooth thickness of a gear.

**Figure 9 polymers-13-03588-f009:**
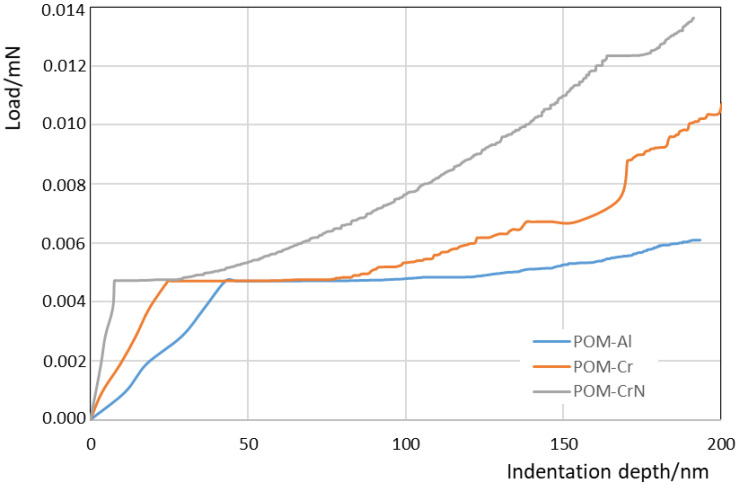
The loading curves obtained during indentations with a very low indentation rate (10^−4^ mN s^−1^).

**Figure 10 polymers-13-03588-f010:**
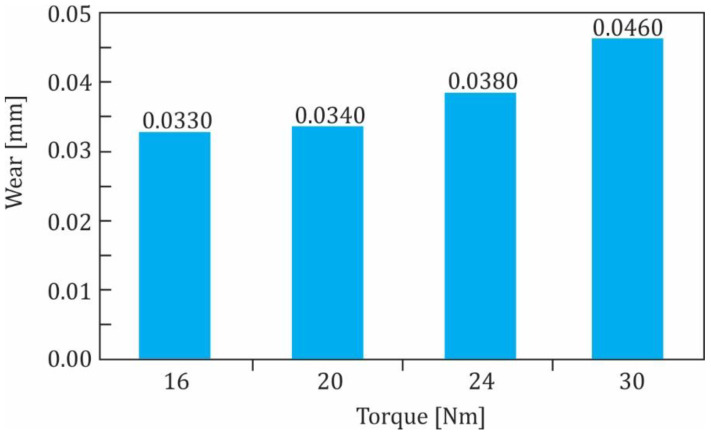
Wear of POM-gears without coating at different torques.

**Figure 11 polymers-13-03588-f011:**
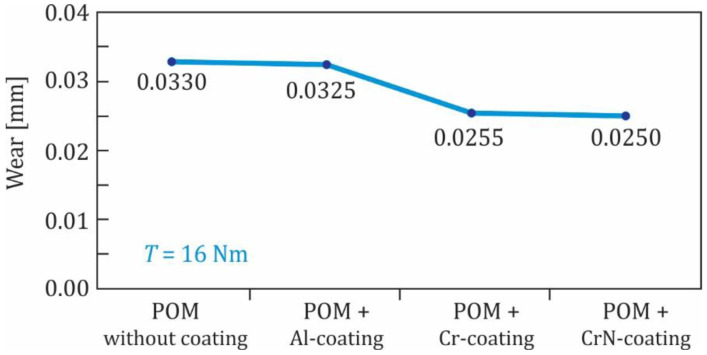
Wear of POM-gears at the torque 16 Nm.

**Figure 12 polymers-13-03588-f012:**
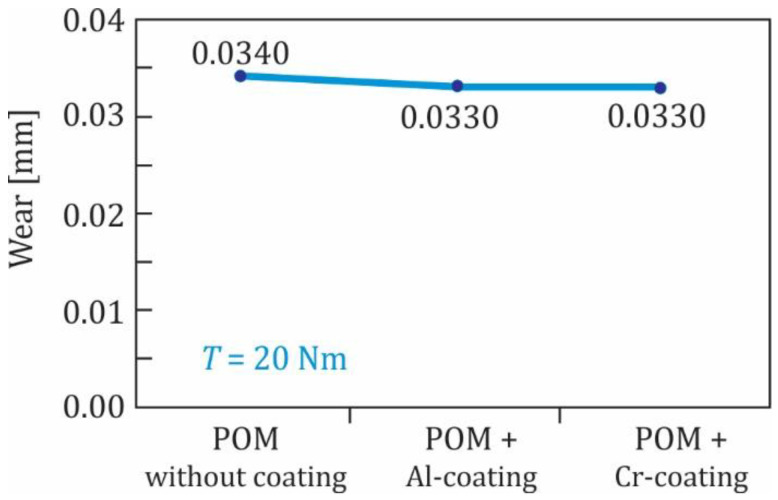
Wear of POM-gears at the torque 20 Nm.

**Figure 13 polymers-13-03588-f013:**
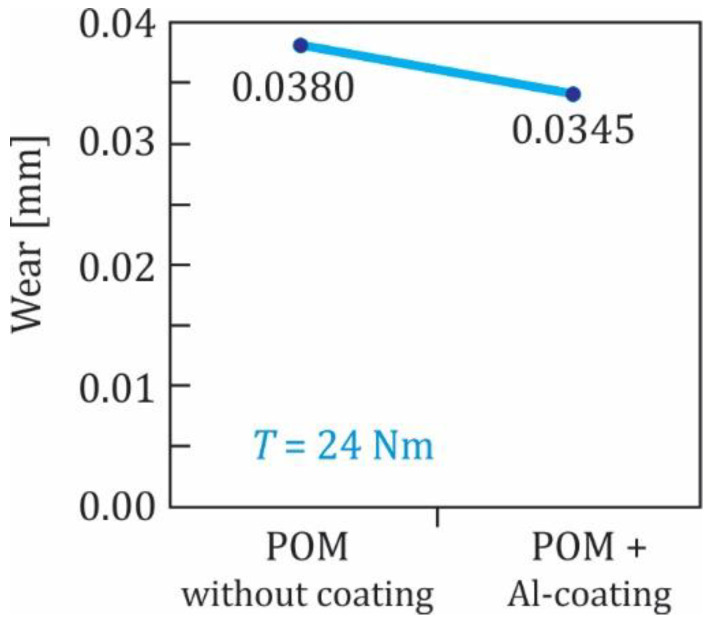
Wear of POM-gears at the torque 24 Nm.

**Figure 14 polymers-13-03588-f014:**
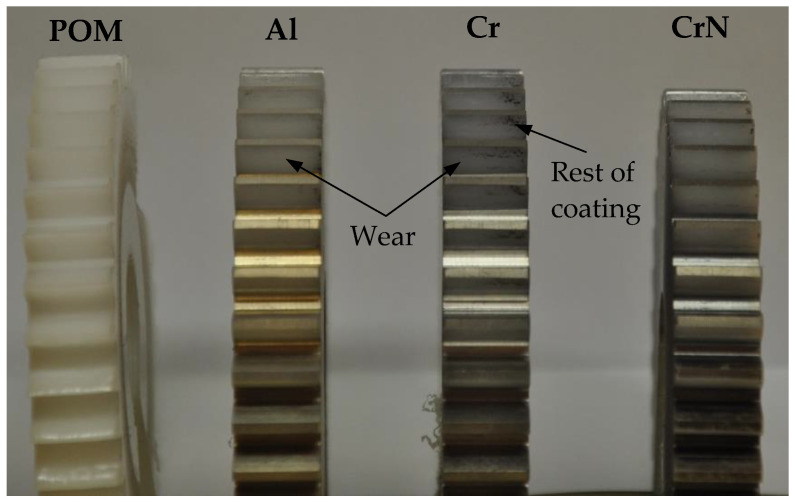
A photograph of the worn gears.

**Table 1 polymers-13-03588-t001:** Process parameters of the analysed coatings.

Coating	Process	Pumping Time (s)	Starting Pressure (mbar)	MFC	Regulation Pressure (mbar)	Process Time (s)	Regulation Energy (kWs)	*T* [°C]	
								min	max
Al	Plasma activation	10	5 × 10^−3^	800	3 × 10^−2^	18	198	500	5000
	Magnetron sputtering	150	4 × 10^−4^	500	2.2 × 10^−3^	62	10,500	30	90
	Plasma polymerisation	1	1.5 × 10^−2^	300	2 × 10^−2^	50	582	500	5000
Cr	Magnetron sputtering	80	6 × 10^−4^	500	3 × 10^−3^	105	10,200	25	90
CrN	Magnetron sputtering	80	6 × 10^−4^	500	3 × 10^−3^	105	10,200	25	90
	Reactive metallisation	90	9 × 10^−4^	120 (190)	3.4 × 10^−3^	67	6200	40	90

**Table 2 polymers-13-03588-t002:** Basic parameters of the tested gear pair.

Parameter	Tested Gear	Supported Gear
Material	POM	Steel (16 MnCr5)
Normal module *m*	2.5 mm	2.5 mm
Pressure angle α_n_	20°	
Helix angle β	0°	
Number of teeth *z*	36	36
Tooth width *b*	14 mm	14 mm
Profile shift coefficient *x*	0	
Centre distance a	90 mm	
Basic rack profile	ISO 53	
Lubrication	Dry (no lubricated)	

**Table 3 polymers-13-03588-t003:** Indentation hardness and modulus in the range between 10 and 100 mN.

	Indentation Hardness (MPa)	Indentation Modulus (GPa)
POM	161.22 ± 2.16	3.50 ± 0.16
POM-Al	269.41 ± 24.67	4.91 ± 0.45
POM-Cr	260.52 ± 8.91	4.76 ± 0.19
POM-CrN	232.67 ± 5.59	4.59 ± 0.34

## Data Availability

The data presented in this study are available on request from the corresponding author.

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
