# Peer review of "Experimental Investigation of the Wear Behaviour of Coated Polymer Gears"

_polymers, 2021, doi:10.3390/polym13203588_

Round 1
Reviewer 1 Report
This paper requires major revision. The comments are given below.
- Authors must include the parameters of the plasma, activation, polymerization and magnetron sputtering
- The surface and cross-sectional SEM micrographs of the Al, Cr or CrN coatings must be included in the manuscript.
- The hardness of the un-coated POM, and hardness of the Al, Cr or CrN coatings must be included in the manuscript.
- The co-efficient of friction of the 4 tribological pairs must be measured and included in the manuscript
- The discussion part must be improved with scientific reasoning
Author Response
Dear Reviewer,
Your comments regarding our original submitted manuscript are very welcome. We have tried our very best to satisfy all the Reviewers’ recommendations and we sincerely hope that the revised manuscript will now be satisfactory. All changes to the original manuscript are marked in blue in the revised manuscript.
Please, find our response on your comments in the attached file "Polymers-1371171_R1-Response to reviewers.docx".

Reviewer 2 Report
Dear Authors,
Congratulations on your work, which is interesting. Please pay attention to some remarks, concerns and suggestions for improvement, as follows:
- In the Abstract, what means "...the metallization of Cr by the magnetron sputtering process"? Is it a common PVD process?
- The results in the Abstract should be improved. It is not enough to report that the tickness of the coatings was not sufficient. Why they failled? How they failled? How they were tested? Abstract must be deeply improved.
- In the Introduction, the PVD process for coating polymer parts is completelly neglected. Thus, I'm recommending the inclusion of a review reference about PVD: doi:10.3390/coatings8110402.
- No support is provided for the use of some coatings, as the presented ones. You need to refer recent experimental works related to the presented coatings, such as: doi:10.3390/coatings11020215; doi: 10.3390/coatings11050555; doi: 10.1016/j.matlet.2020.129187; doi: 10.1016/j.surfcoat.2020.126231; doi: 10.1016/j.wear.2013.01.045.
- The paper is not properly organized. Some information presented in Materials and Methods must be transferred to the Introduction, namely the subsection 2.1.
- In the Experimental Procedure, no information is presented about the targets used, the equipment used for deposition, the deposition conditions, etc. These informations are mandatory to allow a better understanding about what is going on with the remaining tests.
- More explanations are needed about the assembly of the gears for testing, namely the distance and gap between them, pressure of contact, load, and so on.
- Some test conditions are presented in the Results, which is wrong. Please reorganize your paper properly. This information must be organized in the Materials and Methods section.
- Regarding the level of wear reported in Figures 5 and 6, it is clear that PVD coatings cannot solve the problem, because more than 5 um in thickness is too much for a PVD coating.
- Some auxiliary tests should be done in order to understand the level of adhesion of the coatings to the substrates. The plasma activation has produced results? How can we know if no adhesion tests were carried out?
- No images about the thickness analyses are shown. Please show evidences of the coatings produced.
- Please report the main wear mechanisms observed on the worn surfaces and how the coatings suffered wear phenomena.
- The Conclusions...do not conclude anything useful for the surface science. Please deeply improve them, providing useful and transferrable information that can be useful for other scholars and researchers.
To address all the comments and suggestions above is mandatory. Your paper is poor and needs to be deeply improved in order to reach the publication.
Hope this review can help you in understanding what is needed to do to improve the paper.
Kind regards,
Reviewer
Author Response
Dear Reviewer,
Your comments regarding our original submitted manuscript are very welcome. We have tried our very best to satisfy all the Reviewers’ recommendations and we sincerely hope that the revised manuscript will now be satisfactory. All changes to the original manuscript are marked in blue in the revised manuscript.
Please, find our response on your comments in the attached file "Polymers-1371171_R1-Response to reviewers.docx"

Round 2
Reviewer 1 Report
The current revised version of the manuscript submitted by the authors can be accepted for publication
Author Response
Comment:
Moderate English changes required.
Response: The revised version of the manuscript has been examined by Mrs. Shelagh Hedges, our English language proofreader.
Reviewer 2 Report
Dear Authors,
Thank you for addressing my comments and suggestions.
In my opinion, the paper is much better, but there is no Discussion about the results. This aspect needs to be improved. Please pay attention to some papers related to wear of coatings on polymer substrates (just one example: https://doi.org/10.3390/coatings11050555, please look for more in the references of this paper, saving work), to discuss the wear mechanisms involved on your work.
Kind regards,
Reviewer
Author Response
Comment #1: In my opinion, the paper is much better, but there is no Discussion about the results. This aspect needs to be improved. Please pay attention to some papers related to wear of coatings on polymer substrates (just one example: https://doi.org/10.3390/coatings11050555, please look for more in the references of this paper, saving work), to discuss the wear mechanisms involved on your work.
Response: Thank you for this suggestion. In that respect, we accordingly extended the Section “Results and Discussion”. However, some additional investigations regarding the multilayer coatings, scratch tests, measurement of the coefficient of friction etc. will be the subject of our further investigations in this field of research.
Round 3
Reviewer 2 Report
Dear Authors,
My last comments were not addressed. Please take the actions recommended. Please consider this the last chance to do the needed improvements.
Results and Discussion is too poor.
Kind regards,
Reviewer
Author Response
Dear Reviewer,
Thank you for this comment. We are really sorry that our last corrections in the revised version R2 were not satisfactory. As it was pointed out in our last version of the manuscript, the thin surface coatings were removed in a very short time, which was probably due to the poor adhesion of the PVD-coating to the POM substrate. For that reason, some further investigations will need in the future to answer on this question more exactly. Furthermore, the multilayer coatings should also be taken into account. However, we have extended the Section “Results and discussion” in our resubmitted manuscript (version R3) in regard to the other studies you suggested in your comments, and, also, to some aspects regarding the failure characteristics of polymer gears. All changes made in Revision 3 are marked in red.
Yours faithfully,
Corresponding Author: S. Glodež